# Sagnac Loop Based Sensing System for Intrusion Localization Using Machine Learning

Maged A. Esmail [1,*], Jameel Ali [2,3], Esam Almohimmah [2,3], Ahmed Almaiman [2,3], Amr M. Ragheb [2,3] and Saleh Alshebeili [2,3]

1    Communications and Networks Engineering Department and Smart Systems Engineering Laboratory, Faculty of Engineering, Prince Sultan University, Riyadh 11586, Saudi Arabia

2    Electrical Engineering Department, King Saud University (KSU), Riyadh 11421, Saudi Arabia; jhasan.c@ksu.edu.sa (J.A.); ealmohimmah@ksu.edu.sa (E.A.); ahalmaiman@ksu.edu.sa (A.A.); aragheb@ksu.edu.sa (A.M.R.); dsaleh@ksu.edu.sa (S.A.)

3    KACST-TIC in Radio Frequency and Photonics for the e-Society (RFTONICS), King Saud University (KSU), Riyadh 11421, Saudi Arabia

*    Correspondence: mesmail@psu.edu.sa

**Abstract:** Among all optical sensing techniques, the distributed Sagnac loop (SI) sensor has the advantage of being simple to implement with low cost. Most of the proposed techniques for using SI exploit the frequency null method for event localization. However, such a technique suffers from the low spectrum signal power, complicating event localization under environmental noise. In this work, event localization using time-domain instead of frequency null signals is achieved using machine learning (ML), which is increasingly being exploited in many science fields, including sensing applications. First, a training dataset that includes 200 events is generated over a 50 km effective sensing fiber. These time-domain signals are considered as features for training the ML algorithm. Then, the random forest (RF) ML algorithm is used to develop a model for event location prediction. The results show the capability of ML in predicting the event's location with 55 m mean absolute error (MAE). Further, the percentage of test realizations with prediction error > 200 m is 0.7%. The sensing signal bandwidth is investigated, showing better performance results for sensing signals of larger bandwidths. Finally, the proposed model is validated experimentally. The results showed good accuracy with MAE < 100 m.

**Keywords:** Sagnac loop; vibration sensing; machine learning; location prediction

## 1. Introduction

Distributed fiber sensing (DFS) has attracted considerable attraction in different civil and military areas such as perimeter security, pipeline maintenance, and construction health, where DFS provides continuous and live monitoring [1]. Event detection and localization are crucial in sensing applications. However, these represent a difficult task especially in a harsh environment with long distances where the signal-to-noise (SNR) ratio is low, making an event's position localization a challenge. Various fiber-based sensing techniques for event detection and localization have been proposed [2]. In these techniques, the fiber is used as the transmission and sensing medium. Backscattering-based sensing techniques provide a high spatial resolution of several meters for short perimeters. However, for long perimeters, the weak backscattered signal deteriorates the SNR and hence limits the performance of these techniques [3]. In addition, backscattering-based sensing techniques require narrow linewidth lasers and high-speed acquisition devices, which increase the system cost. On the other hand, interferometer-based sensing techniques such as Sagnac and Mach–Zehnder interferometers (MZI) provide good performance in long perimeter applications due to the strong interferometric light offset [4,5]. Sagnac interferometer (SI) loop is a phase modulation-based sensor that exploits optical path

difference [6]. It has the advantage of high SNR, wide frequency range, simple structure, and mature technology. Compared to MZI based sensor, Sagnac has better SNR and uses a light source with low coherence [7–9].

One of the main techniques for event localization in SI is exploitation of the signal's spectrum. Due to the phase difference between the received signals at the photodetector (see Figure 1a), some frequency components disappear, yielding frequency nulls in the spectrum. These nulls correlate to the event position, where a change in the event position yields a change in the distance between the nulls. Hence, such frequency nulls can be used to detect and localize the event. However, searching for these nulls is difficult due to the low power of the frequency spectrum and environmental noise, which renders the spectrum irregular and results in large errors [3,4]. De-noising the nulls using frequency-null curve smoothing techniques could help find the accurate nulls. However, these techniques bear the disadvantage of high complexity [4]. Twice Fast Fourier transform (Twice FFT) was proposed in [4] where a second FFT was applied to the frequency spectrum. The second FFT eliminated the need to focus on the spectrum nulls but focused on the overall distribution. However, this technique suffers from more complexity.

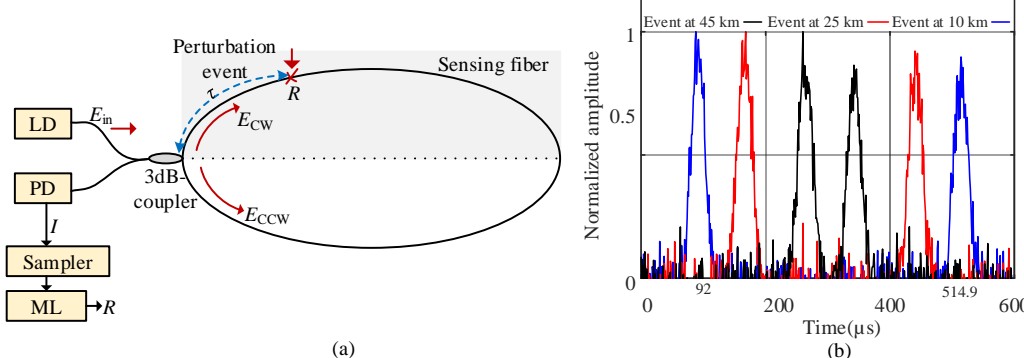

(a)                 (b)

**Figure 1.** (**a**) SI loop distributed sensor, and (**b**) three normalized noisy signals were captured at the PD output due to three different events. The position of the two pulses for every signal is different. LD: laser diode, PD: photodetector, ML: machine learning

Time-delay estimation is another method for event localization in SI loop sensors. In this method, the sensor structure includes more than two paths for the signal. By extracting the phase information from the received signals and performing cross-correlation, the event location could be determined with better accuracy than the frequency null method [9–12]. However, this improvement comes with increased sensor structure complexity and cost.

Machine learning (ML) techniques have gained increased interest and have appeared as a new direction of innovation in many fields [13]. Recently, ML was used to localize the events in SI-based sensors. In [14], a support vector machine (SVM) classifier was exploited to reduce the localization error compared to other traditional methods. The training features of the SVM were the frequency spectrum. The obtained results showed a high classification accuracy greater than 96%. However, this high accuracy was achieved because the test was performed for the same labels that were used for model training. In fact, the model should be tested by a set of data different from the dataset used for training to evaluate its capability in localizing any event that happens at any position over the sensing fiber. This problem was overcome in [15], where a stacking ensemble method combined with two convolutional neural networks (CNNs) were utilized to predict the event location. The signal's frequency spectrum was used as a feature for model training. The results showed a low mean absolute error. However, using the CNN algorithm increased the system complexity. Moreover, the developed model was not validated experimentally.

In [14,15], the null frequency method was exploited as a feature for ML training, which complicated the event location prediction due to the low power of the frequency

spectrum and its sensitivity to environmental noise. In this work, we consider the time-domain output signal of the receiver as input for the ML algorithm. This avoids the null frequency disadvantages and reduces algorithm complexity through the direct use of raw data without pre-processing. In addition, the effective sensing distance in this work is extended to 50 km instead of 8.5 km in [15]. Moreover, the number of realizations generated for training in each event is reduced from 200 as in [15] to 100. This means a 50% reduction in the size of the training dataset. Finally, to validate the proposed model, which is an important step in developing prediction models, laboratory experiments are conducted to generate a test dataset for model validation, which is missed in [15].

In this work, a prediction model based on the random forest (RF) ML is first developed using a numerical simulator. An average mean absolute error (MAE) of 55 m is found. In addition, only 0.7% of the test realizations are found to have an error >200 m. Next, an experiment is conducted to validate the developed numerical model. Three events are generated at three different positions in the sensing fiber. Then, the developed model is used to predict the events' locations. The results indicate that the developed model exhibits good accuracy with MAE < 100 m.

The remainder of the paper is organized as follows. In Section 2, the numerical simulation setup is discussed. The proposed model is studied in Section 3 and the results are presented in Section 4. Lastly, concluding remarks are provided in Section 5.

## 2. SI Loop Sensor Numerical Simulation Setup

Localization sensing requires the construction of a large dataset for ML training and testing. The dataset should include events at different locations and under different conditions, such as from low noise to high noise levels. Such a dataset is particularly difficult to build. The use of customized simulators represent a good alternative. In this work, VPItransmissionMaker, a widely used optical systems commercial simulator, is considered to generate the training and testing datasets. This simulator is often used for conducting numerical simulations in optical communication, where it shows high accuracy comparable to experimental results [16,17]. After ML model development, experiments are carried out to validate the performance of the proposed model. The simulation parameters are set to match the specifications of available equipment in the laboratory.

The structure of the SI loop sensor is illustrated in Figure 1a. In this structure, a 106.245 km single-mode fiber (SMF) is used for building the loop. Half of the loop is used for sensing, and the other half is used as a delay. On the transmitter side, a laser diode (LD) is exploited as a light source. Assume the electric field of the light source is given by

$$E_{in} = E_o \exp j[wt + \theta(t)], \tag{1}$$

where $E_o$ is the electric field's amplitude and $\theta(t)$ is the initial phase. This electric field is divided equally in the 3 dB coupler and then coupled into the SMF. One light signal propagates in the clockwise (CW) direction and the other in the counter-clockwise (CCW) direction of the SMF. When an event occurs, the SMF's refractive index and length change accordingly, inducing phase modulation on each signal. This causes a phase difference between the two light beams. At the coupler output, the two signals interfere before being detected using a photodetector (PD). Assuming an event occurred at a distance $R$ from the coupler, then the two received electric fields at the coupler input are given as

$$E_{ccw} = E_o \exp j[wt + \theta_1(t) + \phi(t - \tau)], \tag{2}$$

$$E_{cw} = E_o \exp j[wt + \theta_2(t) + \phi(t - T + \tau)], \tag{3}$$

where $\tau$ is the propagation time delay between the coupler and location of the event, $T$ is the total propagation delay of the loop, and $\theta_1$ and $\theta_2$ are the initial phases of the CCW and CW signals, respectively. The phase $\phi(t)$ is caused by the induced event. The combined

signals at the output of the 3 dB coupler are applied to the PD, where the signal intensity at the PD output is given by

$$I = (E_{cw} + E_{ccw})(E_{cw} + E_{ccw})^* = 2E_o^2(1 + \cos(\Delta\theta + \Delta\phi)), \qquad (4)$$

where $\Delta\phi = \phi(t - \tau) - \phi(t - T + \tau)$ and $\Delta\theta = \theta_1(t) - \theta_2(t)$.

As an illustration of (4), consider an SI of length 106.245 km. The loop's total propagation delay ($T$) is 520.9 µs assuming the fiber has a refractive index of 1.47. When there is no event, the CW and CCW signals undergo no phase shift, and the output of the PD is a DC signal. If an event occurs at a particular moment at a distance of 10 km from the coupler, a phase change will occur to the CW and CCW signals during this specific moment. The CCW signal will reach the PD after $t = \tau = 49$ µs, while the CW will reach the PD at $t = T - \tau = 471.9$ µs. According to (4), this phase change will cause a change in the signal intensity at the PD output at two different time moments ($t = 49$ µs and $t = 471.9$ µs). Therefore, the signal intensity at the PD output shows two pulses, as illustrated in Figure 1b. Note that the peaks of the pulses are shown at $t = 92$ µs and $t = 514.9$ µs because the event signal (Gaussian pulse) has a particular duration, with a peak at 43 µs. If the event position changes, the two pulses will appear at different positions, as indicated in Figure 1b for events at 25 km and 45 km. Therefore, the output signal of the PD can be used as a feature for the event position during ML training.

The intrusion event is simulated as a Gaussian pulse with 11 kHz bandwidth. The time delay between the two pulses that are indicated in Figure 1b is correlated to the event location. Therefore, it could be exploited as a feature for training ML algorithms.

At the PD output, the signal is sampled at 1 MSa/s. After signal sampling, the ML algorithm is used for event location prediction. It is clear from Figure 1b that as the event becomes closer to the middle of the loop, the two output pulses become closer. For an event's location greater than 50 km, the overlapping between the two pulses complicates the localization objective of the ML since the overlapped pulses produce DC at the PD output. Therefore, the effective fiber length for sensing is assumed to be 50 km. As illustrated in Figure 1b, the added Gaussian white noise affects the signal amplitude and hence complicates the localization process of events. Therefore, powerful techniques are required to accurately determine the event location.

## 3. Machine Learning Model

The development of an ML model consists of two main steps: training and testing. In both steps, a dataset is needed. In this section, dataset generation is first considered followed by discussion about its utilization to train the proposed ML regressor for event localization. The metrics utilized in this study to evaluate the performance of proposed ML localization algorithm are also presented.

### 3.1. Dataset Generation

For ML algorithm training, a dataset is generated using the simulator as follows: A total of 200 events (i.e., disturbances) are generated every 250 m distance in the sensing fiber. After acquiring the data, Gaussian noise is added to each event's realization. First, a total of 100 realizations are generated for each event. Then, random Gaussian noise is added with SNR between 10 dB and 20 dB with 0.1 dB step size. This means a total of $100 \times 200 = 20{,}000$ realizations are generated for training. Every realization includes 600 samples.

To test the performance of the proposed model after the training phase, a test dataset is generated, where one event is generated randomly within each 1 km distance. This means a total of 50 events are generated along the sensing fiber. Figure 2a illustrates how close the test events' positions are to the training events' positions. No test event shares the same position as the training event. The minimum difference distance between a realization from the training dataset and a realization from the testing dataset is 6 m and the maximum difference distance is 125 m. The histogram of the difference distance is

depicted in Figure 2b. The figure indicates that 48% of the test events have a 75 m distance difference to the nearest training events.

For evaluation purposes, 30 noisy realizations are generated. This means there are 1500 (50 × 30) realizations available for model testing. For each test realization, random Gaussian noise is added to produce SNR between 10 dB and 20 dB. Table 1 summarizes the generated dataset for training and testing.

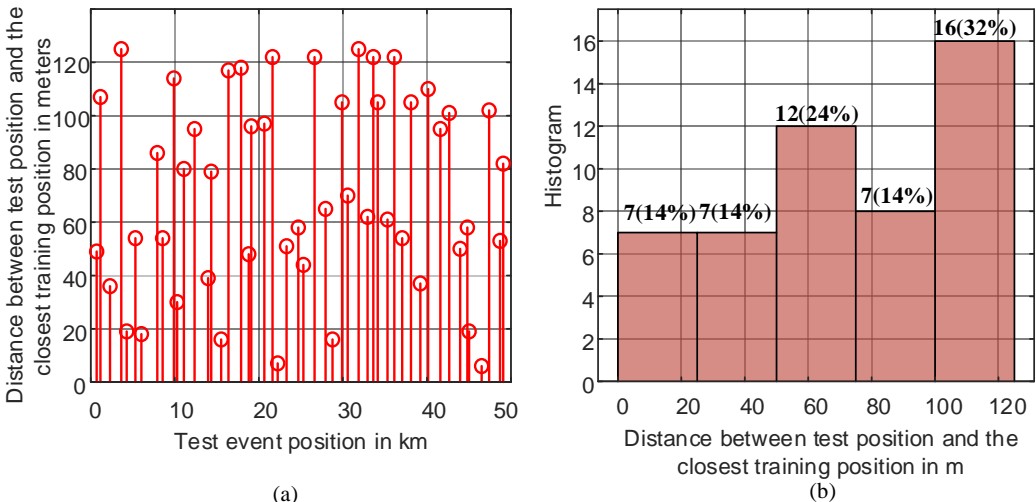

(a)

**Figure 2.** (**a**) Difference between test events' positions and the training events' positions, and (**b**) histogram of the difference distances in Figure 2a.

**Table 1.** Generated dataset for ML training and testing.

| Dataset | Event Location Interval | No. of Locations | Total No. of Realizations |
|---|---|---|---|
| Training | every 250 m | 200 | 20,000 |
| Testing | random in each 1 km segment | 50 | 1500 |

*3.2. Regressor Training*

Before training, the dataset is normalized, where each realization in the dataset is divided by its maximum value. Following this, the training dataset is applied at the input of the ML algorithm. Since the collected raw data from the receiver already include the required features (i.e., the distance between the two pulses, see Figure 1b), there is no need for additional signal processing to extract more features from the training set. This saves time and reduces the complexity of the ML algorithm. In this work, the RF regression model is exploited, which is a supervised ML algorithm that exploits the ensemble learning method to combine predictions from multiple decision tree algorithms for a prediction more accurate than a single decision tree model [18,19]. The bootstrap aggregation method (also called bagging) is used as the ensemble aggregation method. A 10-fold cross-validation is used to ensure there is no overfitting, hence guaranteeing the regressor's reliability of the selected parameters. The best number of regression trees was found to be 50, which yields the minimum mean square error.

*3.3. Performance Metrics*

To evaluate the performance of the proposed ML model, two metrics are used. The first metric is the mean absolute error (MAE), which is defined by [15]

$$MAE = \frac{1}{N} \sum_{i=1}^{N} |y_i - \bar{y}_i| \tag{5}$$

where $N$ is the number of test realizations for each event, $y_i$ is the exact event position, and $\bar{y}_i$ is the corresponding predicted event. The second metric is the percentage of the number of test realizations with a position prediction error larger than a certain distance $X$. This metric (denoted by $\eta$) is defined mathematically as

$$(\eta > X) = (\sum_{i=1}^{N} d_i)/M \times 100, \tag{6}$$

where $M$ is the total number of test realizations, $d_i = 1$ if $|y_i - \bar{y}_i| > X$ and $d_i = 0$ if $|y_i - \bar{y}_i| < X$.

## 4. Results and Discussion

In this section, we first consider the performance of the proposed ML regression model using simulated data. The results of the proposed ML algorithm are also compared with other regressors available in the literature. Further, the effect of event signal type and bandwidth on localization performance are investigated. Experimental results are also provided for validation purposes.

### 4.1. Performance of the Proposed Model

Figure 3a shows the performance of the ML in terms of MAE. In the figure, each of the 50 test events is represented by an identification (ID) number. The results show that the maximum and minimum MAE are 148 m and 28 m, respectively. The average MAE over the 50 events is 55 m. The value of $\eta$ is illustrated in Figure 3b. Almost half of the realizations have a position error > 50 m. The test realizations with a position prediction error > 100 m represent 14.2% of the total number (i.e., 1500). The value of $\eta$ decreases to 0.7% for position prediction error > 200 m.

To look deeper at the MAE of each test location, a boxplot is used for four arbitrary test events, as indicated in Figure 4. A boxplot is a tool that represents the data as a box where the center of the box is the median, whereas the box's bottom edge indicates the 25th percentile and the box's top edge indicates the 75th percentile. From Figure 4, we notice that the box's height is less than 100 m, meaning most of the predicted event's values (30 realizations per event) are close to the median. The standard deviations for the four arbitrary positions in Figure 4 are 41.9 m, 57.7 m, 32.9 m, and 26.9 m. This means, for example, the standard deviation for the 30 prediction values of the position 44.808 km is 41.9 m. The maximum standard deviation for these arbitrary positions is 57.7 m, which reflects the repeatability goodness of the algorithm.

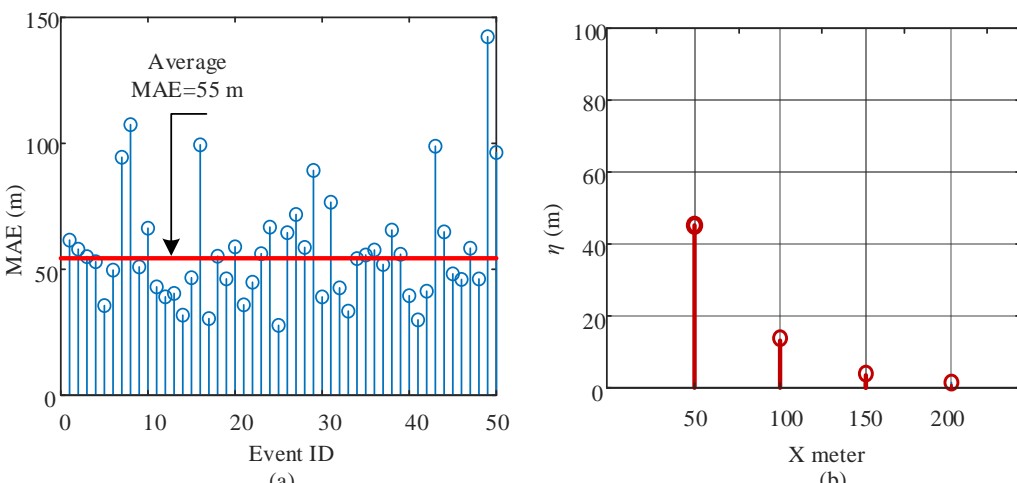

**Figure 3.** (**a**) MAE versus the event position ID, and (**b**) $\eta$ value versus $X$.

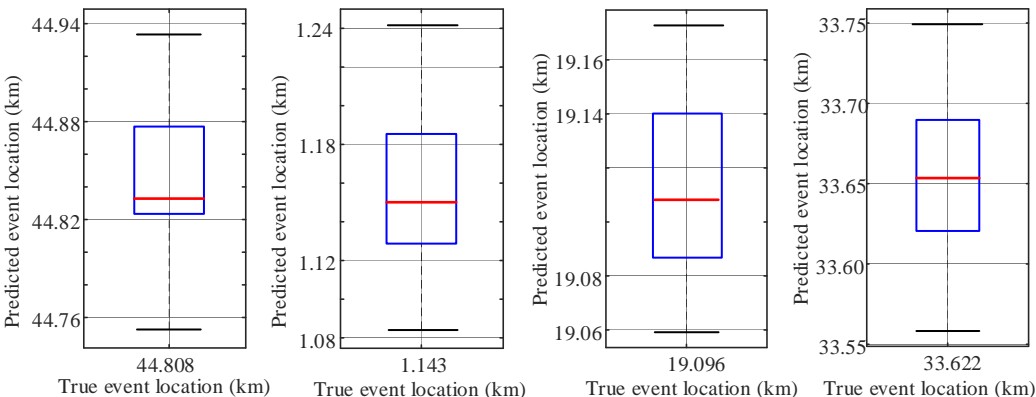

**Figure 4.** Boxplot which shows selected arbitrary predicted events' positions versus true values.

*4.2. Performance Comparison with Other Regressors*

In this subsection, the proposed regressor based on the RF ML is compared with two models that are trained using a single decision tree (DT) regressor and support vector machine (SVM) regressor with a linear kernel. The results for the average MAE and number of test realizations with position prediction error > *X* m are shown in Figure 5a,b, respectively. The average MAE is >500 m for SVM regressor and 188 m for DT regressor, which is high compared to that of the proposed model, i.e., 55 m. Note that for better visualization of the results, the *y*-axis is limited to 500 m in Figure 5a. In addition, the value of $\eta$ for position prediction error > 100 m (200 m) for SVM and DT regressors is 93.8% (88.7%) and 65.2% (38.0%), respectively, which is high compared to that of the proposed model, i.e., 14.3% (0.7%).

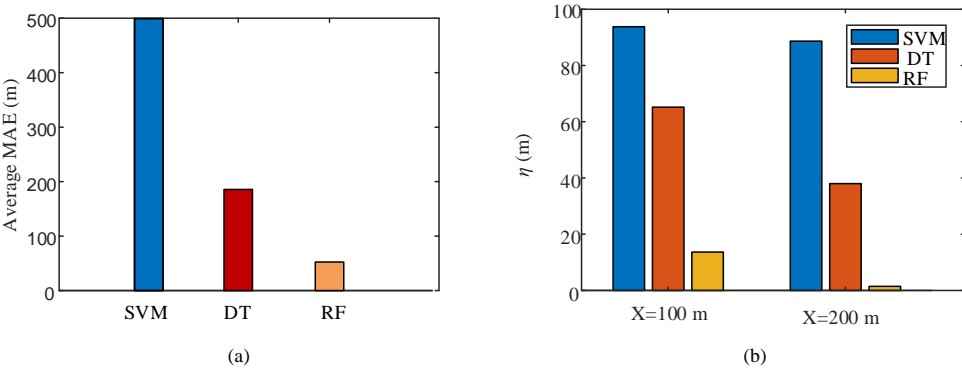

**Figure 5.** Performance comparison of SVM, DT and RF ML algorithms: (**a**) Average MAE, and (**b**) value of $\eta$ with prediction error > *X* m. The *y*-axis in Figure 5a is limited to 500 m for better visualization of the results.

*4.3. Effect of the Sensing Signal Bandwidth*

Next, the performance of the sensing system as a function of the bandwidth of the sensing signal is investigated. Two sensing signals with 11 kHz and 22 kHz bandwidths are considered as indicated in Figure 6a. The average MAE and value of $\eta$ with a prediction error greater than *X* m are illustrated in Figure 6b,c, respectively. In Figure 6b, we see that the higher sensing signal bandwidth has reduced the average MAE by 10.7 m. Similarly, the results in Figure 6c show improvement in the sensing system performance when the sensing signal has larger bandwidth. The value of $\eta$ with a position prediction error larger than 50, 100 m, 150 m, and 200 m has been reduced by 11.2%, 5.9%, 2.2%, and 0.4%, respectively. This improvement can be explained by observing the time duration of the captured sensed signals. Every 250 m, there is an event defined for training the model. If the pulses of the sensed signal (see Figure 6a) have a wide duration (i.e., small bandwidth), the regressor faces difficulty distinguishing between the location of two adjacent events

owing to their high correlation. Using shorter pulses (i.e., large bandwidth) will de-correlate the training realizations leading to better learning.

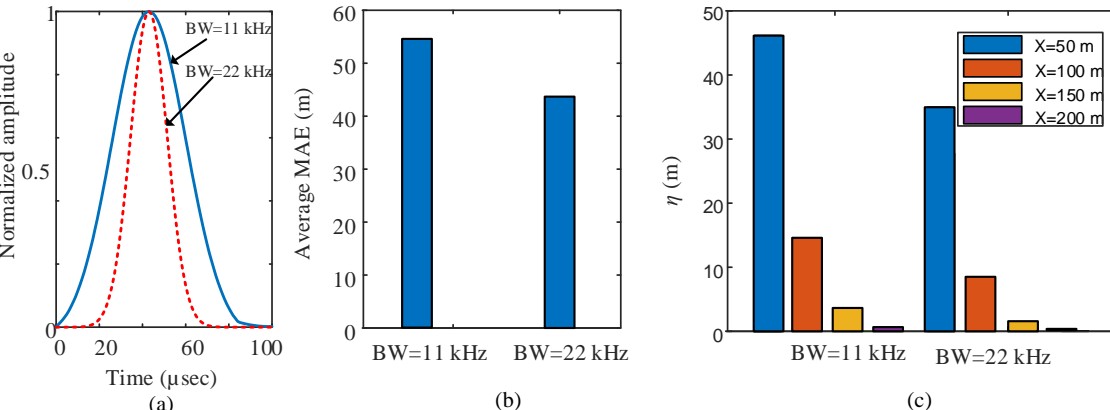

**Figure 6.** Sensing signal bandwidth effect on the system performance: (**a**) Gaussian sensing signals, (**b**) average MAE, and (**c**) value of $\eta$ with prediction error > $X$ m.

### 4.4. Effect of the Event Pulse Type

In the previous subsections, the event was emulated as a Gaussian pulse. This section considers a sinusoidal pulse and investigates the algorithm's performance in localizing the event. A sinusoidal pulse with 11 kHz bandwidth is used for performance comparison with the Gaussian pulse as indicated in Figure 7a. The average MAE and $\eta$ are depicted in Figure 7b,c, respectively, and compared to the results of a Gaussian sensing signal. The results show that the system performance is slightly improved when the sensing signal is a sinusoidal pulse, with a 4 m average MAE better than that achieved by the Gaussian sensing signal. The improvement in performance due to the use of a sinusoidal signal in terms of $\eta$ is 3.5%, 2%, 0.32%, and 0.23% for $X$ = 50 m, 100 m, 150 m, and 200 m, respectively.

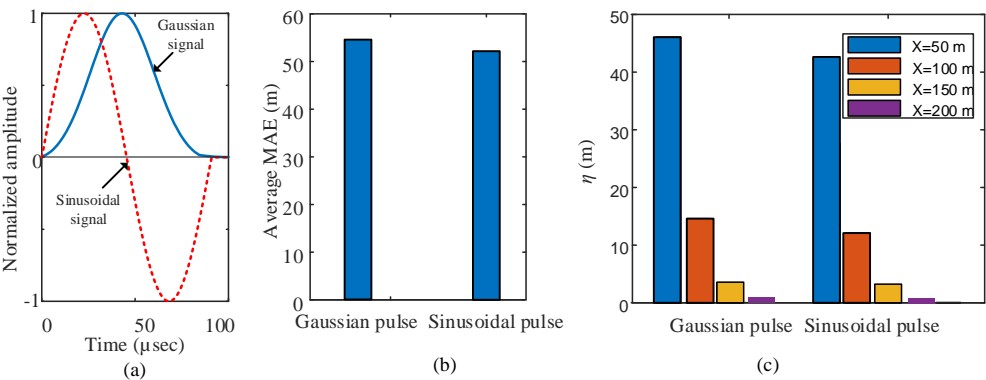

**Figure 7.** Sensing signal type effect on the system performance: (**a**) Sensing signal type, (**b**) average MAE, and (**c**) value of $\eta$ for different values of $X$.

### 4.5. Experimental Results

To prove the validity of the proposed model developed using the simulator, an experiment was conducted to generate multiple sensed signals as indicated in Figure 8a. An Agilent LD (Agilent, N7714A) emitting 10 dBm power at 1550 nm was used as the light source. Multiple SMF spans with different lengths that were available in the laboratory were used to build the loop. The entire loop's length is 106.245 km. A PZT (APC International, 42-1051) device was used to emulate the intruder signal. A Gaussian electrical pulse was generated using an arbitrary waveform generator (PeakTeck, AWG 4125) with amplitude 2.4 Vp-p and 11 kHz bandwidth and applied to drive the PZT in order to emulate the event. A block of soundproof foam was used to keep the PZT from having

an effect on the whole sensing fiber. A 10 m length of the SMF was wrapped around the PZT. A polarization controller was used in the loop to manipulate the polarization effect. The PZT was placed at three different locations in the loop: Loc1 = 14.190 km, Loc2 = 24.837 km, and Loc3 = 43.345 km. These locations were measured using an OTDR device. A photoreceiver (Optilab, LR-12.5GR) was used to receive the optical signal, and its output signal was sampled using an oscilloscope (Keysight, DSO-X 3034A) with 1 MSa/s sampling speed. The sampled signal was applied to the developed model input to predict the event location.

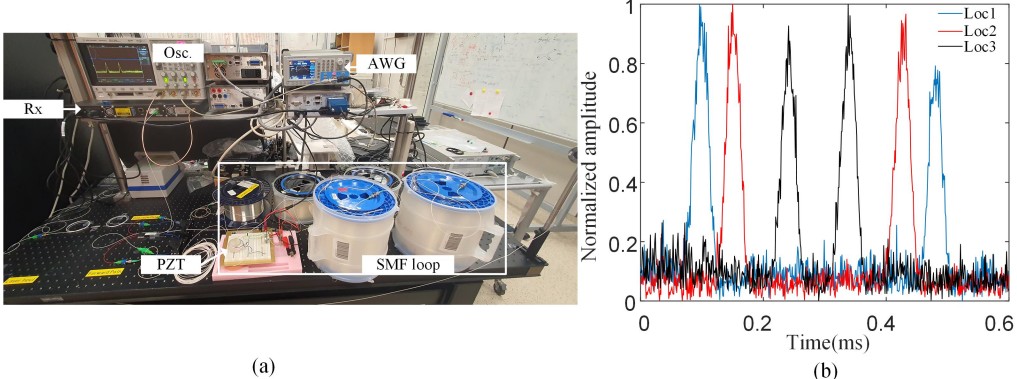

(a)  (b)

**Figure 8.** (**a**) Experimental setup, and (**b**) received sensed signals of the three defined events.

Figure 8b shows three realizations captured by the oscilloscope, which represent three events located at the three predefined locations. The realizations generated from the experiments were applied to the prediction model that was developed by the simulations. To calculate the prediction error, five measurements were taken for each location, and the average MAE was calculated. The predicted location for each measurement and the average MAE are listed in Table 2. The results show that the average MAE error is close to the simulation value (55 m) for location 1 and location 2. For location 3, it is higher than the simulation-based value, but still acceptable. Similarly, Table 2 shows the standard deviation (std) of the prediction error for each location. The maximum and minimum std are 36.5 m and 25.3 m for Loc2 and Loc3, respectively. Note that the proposed model is capable of localizing single events only. Localizing simultaneous events requires the development of another model, which is outside the scope of this work.

**Table 2.** Experimental prediction results.

| | Predicted Location (km) | | | | | Exact Loc. (km) | Avg. MAE ± Std (m) |
|---|---|---|---|---|---|---|---|
| Loc1 | 14.116 | 14.070 | 14.205 | 14.119 | 14.097 | 14.190 | 74.6 ± 34.5 |
| Loc2 | 24.941 | 24.844 | 24.929 | 24.872 | 24.759 | 24.837 | 63.4 ± 36.5 |
| Loc3 | 43.258 | 43.250 | 43.220 | 43.285 | 43.217 | 43.345 | 99.1 ± 25.3 |

## 5. Conclusions

In this work, ML was exploited for event localization in an SI-based sensor structure. The numerical simulation results indicated the feasibility of ML to predict the event location with an average MAE of 55 m in a 50 km sensing fiber. The investigation showed that an event with high bandwidth would be predicted with less location error. The performance of the proposed RF-based model is compared with SVM and DT based-models. The results revealed that the RF model outperforms both models. The developed model was validated experimentally in a laboratory setup. The results showed that the model was capable of predicting an event's location with low prediction error. As future work, we consider developing an ML-based model for simultaneous event localization using the same simple SI sensor structure.

**Author Contributions:** Conceptualization, M.A.E., S.A. and A.M.R.; methodology, M.A.E., A.A., J.A. and S.A.; software, M.A.E. and J.A.; validation, M.A.E., J.A. and E.A.; formal analysis, M.A.E., A.A. and S.A.; investigation, M.A.E., A.A., A.M.R. and J.A.; writing—original draft preparation, M.A.E.; writing—review and editing, M.A.E., S.A., A.M.R. and E.A.; supervision, M.A.E. and S.A.; project administration, M.A.E., S.A. and A.M.R.; funding acquisition, S.A. All authors have read and agreed to the published version of the manuscript.

**Funding:** This work was supported by the National Plan for Science, Technology, and Innovation (MAARIFAH), King Abdulaziz City for Science and Technology, Kingdom of Saudi Arabia, Award Number 3-17-09-001-0012.

**Institutional Review Board Statement:** Not applicable.

**Informed Consent Statement:** Not applicable.

**Data Availability Statement:** Not applicable.

**Conflicts of Interest:** The authors declare no conflict of interest. The founders had no role in the design of the study; in the collection, analyses, or interpretation of data; in the writing of the manuscript, or in the decision to publish the results.

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
