# Peer review of "Sagnac Loop Based Sensing System for Intrusion Localization Using Machine Learning"

_photonics, doi:10.3390/photonics9050275_

Round 1
Reviewer 1 Report
The authors have made corrections according to most of the comments formulated by the reviewers at the previous review round. The experimental validation of the proposed model was carried out. My suggestion is that the manuscript can be published without further ammendments.
As a suggestion to the authors for their further work on the topic, a thought comes to mind that the dataset for ML model training can include perturbations of various types (gaussian pulses of different bandwidths, sin waves, etc.), which may lead to better ability of the model to generalize.
Author Response
The authors would like thank the reviewer for the effort in evaluating the manuscript. Our response to the reviewer's comments is attached.

Reviewer 2 Report
The object of research described in the manuscript is application of Machine Learning (ML) techniques in a distributed optical fiber intrusion sensor using a Sagnac interferometer.
Distributed optical fiber sensors using a Sagnac interferometer have been an active area of research for about three decades with several research groups actively taking part. These sensors have a wide range of present and potential applications, which makes them interesting to several industries. Therefore, the manuscript is of considerable interest to a relatively large audience. However, some issues should be addressed prior to publication of this manuscript.
First, Section 4.5 is written in Simple Present tense, with occasional use of Future (e.g. ‘The realizations generated from the experiments will be applied to the prediction model’ - lines 242-243). Unless decided otherwise by the Editorial Staff, this section should be rewritten using Past tenses.
Second, there is a number of errors in the language of the manuscript. In particular, at line 46 the text should read: ‘FFT was applied to the frequency spectrum’. There is most probably a missing word in the phrase ‘a widely commercial simulator’ (lines 93-4). In other places quality of language should be improved, by e.g. use of ‘selected’ instead of ‘some’ in the caption of Figure 4 or more straightforward language, e.g. ‘proposed RF-based model’ (line 256) instead of ‘proposed model (i.e., RF-based model)’.
Finally, typesetting errors can be encountered in the manuscript. For example, ‘occur ed’ (line 106) or ‘to derive the PZT’ (line 232)
In conclusion, the manuscript cannot be published in its present form, before problems outlined in this review are addressed.
Author Response

(The authors gave the same response as above.)

Reviewer 3 Report
The authors demonstrated a new method for Sagnac Loop Based Sensing System for Intrusion localization using Machine Learning. In this work, ML has been exploited for event localization in SI-based sensor structure.The performance of the proposed RF-based model is compared with SVM and DT based-models. The results showed that the RF model outperforms both models. The paper makes a good contribution in terms of application in the field. Yet, some method and data descriptions are not clear in the manuscript. I recommend publication, if the following mandatory revisions are made well. Below are some specific comments:
- Please add more relevant recent references (2019-2021) in the introduction, especially those related to distributed Sagnac loop (SI) sensor and interferometer-based sensing techniques such as Sagnac, Mach-Zehnder interferometer (MZI) and Machine learning (ML) techniques.
- The systems and events (signal response curves) of Figure 1(a) and (b) should be clearly explained in the text. For example, move page 3 The structure of the SI loop sensor is illustrated in Fig. 1(a). In this structure, 106.245 km single-mode fiber (SMF) is used for building the loop. .......Move to page 2 pages and 36 lines.
- From line 100-124, the text is confusing.
Please describe the curves in the Fig. 1(b) and in the main text. - Page 6 line 186, meaning most of the predicted event’s values (30 realizations per event) are close to the median. What is the repeatability of the measurements?
- The measurement results in this paper, the measurement values must be presented strictly. Page 9, line 244, To calculate the prediction error, five measurements are taken for each location, and the average MAE is calculated. What is the standard deviation of this measurement? Such as page 10, Table 2., Avg. MAE (m) ± sd., 47.9± sd...
Author Response

(The authors gave the same response as above.)
